# The HIF-1α/EGF/EGFR Signaling Pathway Facilitates the Proliferation of Yak Alveolar Type II Epithelial Cells in Hypoxic Conditions

**DOI:** 10.3390/ijms25031442

**Published:** 2024-01-24

**Authors:** Biao Wang, Junfeng He, Yan Cui, Sijiu Yu, Huizhu Zhang, Pengqiang Wei, Qian Zhang

**Affiliations:** 1Laboratory of Animal Anatomy & Tissue Embryology, Department of Basic Veterinary Medicine, Faculty of Veterinary Medicine, Gansu Agricultural University, Lanzhou 730070, China; wangb0503@163.com (B.W.); hejf@gsau.edu.cn (J.H.); sijiuy@126.com (S.Y.); alanbel0831@163.com (H.Z.); wei13140077@163.com (P.W.); zq880204@126.com (Q.Z.); 2Gansu Province Livestock Embryo Engineering Research Center, Department of Clinical Veterinary Medicine, Faculty of Veterinary Medicine, Gansu Agricultural University, Lanzhou 730070, China

**Keywords:** hypoxia, alveolar epithelial cells, proliferation, apoptosis, yak

## Abstract

The yak is a unique creature that thrives in low-oxygen environments, showcasing its adaptability to high-altitude settings with limited oxygen availability due to its unique respiratory system. However, the impact of hypoxia on alveolar type II (AT2) epithelial cell proliferation in yaks remains unexplored. In this study, we investigated the effects of different altitudes on 6-month-old yaks and found an increase in alveolar septa thickness and AT2 cell count in a high-altitude environment characterized by hypoxia. This was accompanied by elevated levels of hypoxia-inducible factor-1α (HIF-1α) and epidermal growth factor receptor (EGFR) expression. Additionally, we observed a significant rise in Ki67-positive cells and apoptotic lung epithelial cells among yaks inhabiting higher altitudes. Our in vitro experiments demonstrated that exposure to hypoxia activated HIF-1α, EGF, and EGFR expression leading to increased proliferation rates among yak AT2 cells. Under normal oxygen conditions, activation of HIF-1α enhanced EGF/EGFR expressions which subsequently stimulated AT2 cell proliferation. Furthermore, activation of EGFR expression under normoxic conditions further promoted AT2 cell proliferation while simultaneously suppressing apoptosis. Conversely, inhibition of EGFR expression under hypoxic conditions had contrasting effects. In summary, hypoxia triggers the proliferation of yak AT2 cells via activation facilitated by the HIF-1α/EGF/EGFR signaling cascade.

## 1. Introduction

Yaks inhabit the hypoxic Tibetan Plateau at an average altitude of ≥3000 m, rendering them a natural and favorable model for studying hypoxic adaptation. The lungs serve as the primary respiratory organ responsible for gas exchange during oxygen and carbon dioxide respiration. Due to its direct exposure to air, the alveolar epithelium exhibits a high degree of sensitivity toward fluctuations in environmental oxygen levels, which can trigger a cascade of biochemical reactions [1]. The alveolar epithelium contains two cell types: type 1 (AT1) and type 2 (AT2). AT1 cells constitute an essential component of the blood–air barrier, yet they lack proliferative capacity [2]. AT2 cells, which are regional lung epithelial progenitor cells that proliferate during lung development to generate alveolar structures, are mobilized after lung injury to repopulate alveoli [3]. When AT1 cells are damaged, adjacent AT2 cells are stimulated to proliferate and transform into AT1 cells [4]. In addition, AT2 cells secrete surfactant-associated proteins (SP), which play a crucial role in maintaining the alveolar microenvironment [5]. Perturbations in the functional state of AT2 have been implicated in the pathogenesis of various pulmonary diseases. However, there is currently no investigation on the regulation of proliferation and apoptosis in yak AT2 cells surviving under hypoxic conditions.

Hypoxia-induced factors (HIFs) are key regulators of the hypoxic response [6]. Under hypoxic conditions, the expression of numerous genes is regulated by HIFs to regulate cell apoptosis, proliferation [7], and transformation [8]. In addition, the epidermal growth factor (EGF) is highly expressed under hypoxic conditions and regulates cell growth through the EGFR signaling pathway. HIF-1α induces EGF expression [9] and subsequently activates the EGFR-PI3K/AKT signaling pathway to promote cell growth under hypoxic conditions [10]. Studies have demonstrated that lung hypoxia activates the HIF-1α signaling pathway to facilitate AT2 proliferation and differentiation for tissue repair [11]. However, it remains unclear how the HIF-1α/EGF/EGFR axis modulates AT2 cell proliferation in hypoxic environments, particularly in yaks—large mammals that inhabit high-altitude regions throughout the year.

In this study, we investigated the effects of hypoxia on the production of HIF-1α/EGF/EGFR axis-related factors in the lungs of yaks and the regulation of AT2 cell proliferation and apoptosis in hypoxia. This study aims to lay the foundation for understanding hypoxia adaptation in yaks.

## 2. Results

### 2.1. Alteration of the Alveolar Histologic Structure and Expression of HIF-1α and EGFR in 6-Month-Old Yaks by Hypoxia

The hematoxylin and eosin (H&E) staining results revealed a significant thickening of the interalveolar septum in 6-month-old yaks (HY) that exhibited survival at high altitudes (Figure 1A,B). In addition, we detected the effect of hypoxia on the apoptosis of alveolar epithelial cells. TUNEL staining revealed a higher rate of apoptosis of alveolar epithelial cells in the HY group (Figure 1C,D). Subsequently, immunofluorescence and Western blotting showed that HIF-1α was mainly distributed in the bronchial and alveolar epithelial cells of six-month-old yak lungs (Figure 1E), and hypoxia could promote the expression of HIF-1α and EGFR (Figure 1F–H).

### 2.2. Hypoxia Alters AT2 Cell Proliferation and Alveolar Surfactant-Associated Protein (SP) Secretion in 6-Month-Old Yaks

The immunohistochemistry results demonstrated specific expression of the SPC protein in AT2 cells (Figure 2A). Subsequently, we quantified the number of SPC- and Ki67-positive cells using immunofluorescence. Our findings revealed that Ki67 was predominantly localized in pulmonary alveolar epithelial cells, with both SPC- and Ki67-positive cells being higher in the HY group than in the LY group (Figure 2B–E). These observations suggest a potential role of hypoxia in promoting AT2 cell proliferation. Additionally, analysis of secreted proteins from AT2 cells revealed significantly elevated levels of SPA and SPC in the HY group relative to the LY group, while no significant differences were observed for SPB and SPD expression (Figure 2F–H).

### 2.3. Primary Culture and Identification of Yak AT2 Cells

To investigate the role of hypoxia in AT2 cell proliferation in vitro, primary AT2 cells were isolated and cultured from yak lungs. As shown in Figure 3A,B, the cultured cells exhibited adhesion and extension on the culture plate within 72 h of initial culture, followed by a paving stone-like appearance upon confluence (on days 6–7). Identification using transmission electron microscopy confirmed their identity as AT2 cells (Figure 3C,D), further supported by the expression of the specific markers SPC for AT2 cells and Aqp-5 for AT1 cells (Figure 3E). The results demonstrated that over 95% of the isolated cells were indeed AT2 cells.

### 2.4. Hypoxia Promotes AT2 Cell Proliferation In Vitro

To mimic the living environment of AT2 cells in vivo, we conducted an air–liquid culture of AT2 cells (Figure 4A). Subsequently, we evaluated the protein expression levels of HIF-1α, EGF, and EGFR in AT2 cells exposed to hypoxia for varying durations (12, 24, 36, 48, and 60 h), revealing a time-dependent upregulation of HIF-1α, EGF, and EGFR protein levels upon exposure to 5% O_2_ (Figure 4B–E). According to the STRING database search, HIF-1α exhibits a targeting relationship with various factors within the EGFR signaling pathway, thereby enabling regulatory control over the expression of EGF and EGFR (Figure 4F). The impact of hypoxia on AT2 cell proliferation was assessed by examining the protein expression levels of the proliferation markers Ki67 and PCNA. The results of Western blotting and immunofluorescence demonstrated a time-dependent up-regulation in the expression of PCNA protein and an increase in Ki67 cell positivity upon exposure to 5% O_2_ (Figure 4G–J).

### 2.5. Activation of HIF-1α Promotes the Proliferation of Yak AT2 Cells

To investigate the impact of HIF-1α on yak AT2 cell proliferation, we utilized ML228 to activate HIF-1α under a 21% O_2_ concentration and evaluated the proliferative capacity of yak AT2 cells. Cell viability assays confirmed that treatment with ML228 at concentrations ranging from 5 to 10 μM significantly enhanced AT2 activity, while concentrations exceeding 20 μM exhibited cytotoxic effects (Figure 5A,B). Western blot analysis demonstrated successful elevation of HIF-1α/EGF/EGFR protein levels following ML228 treatment (Figure 5C–F). Subsequently, we assessed the proliferative potential of AT2 cells by examining PCNA protein expression levels and the number of Ki67-positive cells. The results revealed that ML228 upregulated both PCNA protein level and Ki67-positive cell count (Figure 5C,G–I).

### 2.6. EGFR Pathway Promotes Proliferation and Inhibits Apoptosis in AT2 Cells under Hypoxia

To investigate the role of EGFR in hypoxic proliferation of AT2 cells, we utilized AG1478 and NSC228155 to inhibit and activate EGFR under hypoxia and normoxia conditions, respectively. Western blot analysis revealed that inhibition of EGFR under 5% O_2_ significantly decreased the expression levels of PCNA and Bcl-2 proteins, while increasing the expression level of Bax protein. Conversely, activation of EGFR resulted in a significant upregulation in PCNA and Bcl-2 protein expression levels, accompanied by a downregulation in Bax protein expression levels under 21% O_2_ (Figure 6A–E). Furthermore, Edu assay confirmed that activation of EGFR promoted yak AT2 cell proliferation (Figure 6F,G).

## 3. Discussion

This study provides robust theoretical support for investigating hypoxia adaptation in yaks. By conducting a comparative analysis of the histological structures of lungs from one-month-old yaks surviving at varying altitudes, we observed thickening of alveolar septa and an increase in the number of AT2 cells. Moreover, there was a significant elevation in the proliferation and apoptosis rate of alveolar epithelial cells within the high-altitude group, suggesting that hypoxia may stimulate lung development in yaks. Furthermore, our findings demonstrate that hypoxia induces AT2 cell proliferation in yaks through activation of the HIF-1α/EGF/EGFR signaling pathway. This research offers valuable insights into the mechanisms underlying environmental adaptation employed by yaks.

The morphological findings regarding the hypoxic group indicate thickening of the alveolar septum, which is consistent with previous reports in Tibetan pigs [12]. As indicated, these physiological changes are adaptations to cope with reduced oxygen availability. The alveolar septum, composed of a thin layer of connective tissue with dense capillaries and abundant fibers, is covered by flat alveolar cells [13]. Previous studies have demonstrated that yak lungs possess a greater abundance of elastic fibers [14] and a denser network of blood vessels compared to other cattle species [15]. With the thickening of the alveolar septum, there is also an increase in the number of capillaries and elastic fibers [16]. These elastic fibers provide robust retraction force for the yaks to facilitate air exchange between the external atmosphere and blood within their lungs [17]. Additionally, the increased number of AT2 cells may contribute to enhanced lung function through their involvement in surfactant production and repair mechanisms [18,19].

In this study, we also observed an enhanced proliferation of AT2 cells in the lungs of yaks exposed to a hypoxic environment, accompanied by an increased level of apoptosis. These findings suggest that hypoxia exposure may stimulate lung development and regeneration processes in yaks. The process of apoptosis, which is under genetic control and involves the activation, expression, and regulation of a cascade of genes, represents an active phenomenon rather than a passive one. In contrast to cell necrosis, apoptosis is not a pathological self-injury but rather an intentional programmed cell death aimed at enhancing adaptation to the surrounding environment. The yak lung demonstrates adaptability to hypoxic stress by regulating the proliferation and apoptosis of AT2 cells. Previous research has shown that under hypoxic conditions, cells tend to undergo apoptosis rather than proliferation [20]. Apoptosis serves as a prerequisite for cell proliferation [21]. The heightened proliferative capacity of AT2 cells may be attributed to their activity of replenishing damaged or diseased cells and maintaining normal tissue structure and function. Conversely, the elevated level of apoptosis likely functions in eliminating damaged or unwanted cells. This study provides significant information in comprehending how animals adapt to extreme environmental conditions. Further exploration into the underlying mechanisms will shed light on self-regulation and adaptation strategies employed by organisms when faced with challenges, offering novel insights into human health issues.

Our study revealed that the proliferation of yak AT2 cells induced by hypoxia was mediated through the activation of the HIF-1α/EGF/EGFR signaling pathway. It is well known that HIF-1α, a protein primarily regulating oxygen receptors, becomes activated under hypoxic conditions and participates in various physiological responses [9]. EGF, a cytokine binding to EGFR, initiates multiple signaling pathways involved in physiological processes such as cell proliferation, survival, and apoptosis [22]. Through our investigation, we observed a significant upregulation of HIF-1α expression in yak lung tissues exposed to a hypoxic environment. Additionally, the expression levels of EGF and EGFR were also significantly enhanced. These findings suggest that the adaptability of yak lungs to hypoxic environments may be associated with the activation of the HIF-1α/EGF/EGFR axis signaling pathway. In vitro experiments further demonstrated an improved proliferative ability of AT2 cells from yak lungs under hypoxic conditions. Subsequent studies revealed that two key factors, namely HIF-1α and EGFR, play crucial roles in regulating AT2 cell proliferation within a hypoxic environment. Inhibition of either HIF-1α or EGFR expression resulted in a decreased proliferative ability among AT2 cells. Specifically, the interaction between HIF-1α and EGF is prompted by the hypoxic environment [23], which subsequently activates the EGFR signaling pathway [24]. This synergistic activation promotes Yak AT2 cell proliferation through multiple pathways. Firstly, HIF-1α can directly regulate gene transcription and translation processes leading to increased synthesis of related proteins; it can also indirectly affect other signaling pathways such as PI3K/AKT and MAPK [25,26,27], ultimately stimulating DNA synthesis and mitosis within AT2 cells [28]. Furthermore, under hypoxic conditions, EGF is implicated in the regulation of cell proliferation through its binding to the EGFR receptor [29]. The EGF/EGFR signaling pathway represents a classical family of growth factor receptors that exert pivotal roles in numerous physiological and pathological contexts [30]. In the presence of a hypoxic stimulus, AT2 cells need to repair and regenerate faster to adapt to changes in the external environment due to the lack of sufficient oxygen supply to the respiratory system. EGF can initiate a series of cascading reactions by binding to EGFR, which promotes the AT2 cells to synthesize DNA at a higher rate and complete mitosis.

In conclusion, this study demonstrates that 6-month-old yaks residing in hypoxic environments exhibit the adaptive characteristic of thickened alveolar septa, which effectively facilitates pulmonary gas exchange and ensures their survival. Furthermore, our investigation reveals the involvement of the HIF-1α/EGF/EGFR axis in the adaptation process of yak lungs to hypoxia within such environments, regulating both proliferation and apoptosis of AT2 cells. These findings provide a crucial foundation for further comprehension of the mechanisms underlying hypoxia adaptation in highland animals and hold potential implications for treating hypoxia-related diseases.

## 4. Materials and Methods

### 4.1. Animals and Sample Collection

Lung tissues were collected from 6-month-old yaks (*n* = 3 + 3) residing at altitudes of 2500 m and 4500 m in the Qinghai area, respectively. These yaks had been exposed to distinct altitudinal conditions for an extended duration. A portion of the tissue samples was fixed in a solution containing 4% paraformaldehyde (PFA) in phosphate-buffered saline (PBS, pH 7.4) for immunofluorescence analysis, while the remaining samples were immediately frozen in liquid nitrogen for molecular biological analysis. The promoted physical examination and serum chemistry analyses yielded normal results. All experimental procedures were conducted following the guidelines set by the Ministry of Science and Technology of the People’s Republic of China (approval number: 2006-398). This study was approved by the Animal Ethics Committee of Gansu Agricultural University.

### 4.2. Immunofluorescence Assays

Lung tissue was fixed with 4% PFA for at least 48 h, and then dehydrated and embedded in paraffin. The embedded lung tissue was sectioned into slices with a thickness of 4 μm and subsequently affixed onto slides for staining. AT2 cells were fixed with 4% paraformaldehyde for 30 min at room temperature and then permeabilized with 0.5% Tryon-100 for 30 min. Cells washed with PBS were blocked with 5% bovine serum albumin (BSA) (SW3015, Solarbio, Beijing, China) for 1 h at room temperature.

Immunofluorescence staining was performed using primary antibodies against SPC (1:200, AP53886PU-N, OriGene, Maryland, MD, USA), Ki67 (1:200, GTX00538, Genetex, Texas, USA), or HIF-1α (1:100, AF1009, Affinity, Changzhou, China). Following overnight incubation at 4 °C, the slides were washed and subsequently incubated with the corresponding secondary antibody, IgG Alexa Fluor 594 (1:1000, 8889 s, Cell Signaling Technology, Boston, MA, USA), for 1 h at 37 °C in a light-restricted environment. This was followed by a 5 min incubation with 4′,6-diamidino-2-phenylindole (DAPI, 10 μg/mL) in darkness. Images were promptly captured using a fluorescence microscope.

### 4.3. TUNEL Staining

The fixed tissues were sectioned according to the TUNEL kit (45197300, Roche Group, Basel, Switzerland). For subsequent procedures, the sections were dewaxed with water and treated with Proteinase K working solution at 37 °C for 25 min. After cleaning, the TUNEL reaction mixture was prepared by combining 50 μL of Enzyme Solution with 450 μL of Label Solution. The reaction was carried out in a dark wet box at 37 °C for 1 h. Subsequently, 50 μL of Converter-POD was added to the sample and incubated in a dark wet box at 37 °C for an additional 30 min. After cleaning, DAB chromogenic agent (50–100 μL) was applied and the reaction was observed at room temperature for 10 min. Hematoxylin staining followed briefly before rinsing with running water. Alcohol dehydration, xylene transparency, and sealing with neutral gum were performed accordingly. Finally, microscopic observation was conducted using Image J ((2.6.1.0)) software to count the number of apoptotic and normal nuclei and calculate the percentage of apoptotic cells with positive expression.

### 4.4. RNA Isolation and Quantitative Real-Time PCR (RT-qPCR)

Total RNA was isolated from lung tissue samples using TRIzol reagent (Solarbio, Beijing, China). The concentration and purity of extracted RNA were detected by spectrophotometer. The extracted RNA was reverse transcribed into cDNA using an Evo M-MLV Reverse Transcription Kit (Accurate Biology, Changsha, China). SYBRR Premix Ex TaqTM II (TaKaRa, Beijing, China) was used for quantitative real-time fluorescence analysis. The primers were synthesized by Shanghai Sangong Biological Company (Table 1). mRNA expression was calculated by the 2^−ΔΔCt^ method, and β-actin was the internal reference gene.

### 4.5. Protein Extraction and Western Blotting

By adding 1 mL of RIPA buffer and 10 µL of benzylmethylsulfonyl fluoride (Solarbio, Beijing, China), total protein was extracted. The quantification of all proteins was performed using an enhanced BCA protein assay kit (Thermo Fisher Scientific, Niederelbert, Germany). Following denaturation, proteins were separated by sodium dodecyl sulfate-polyacrylamide gel electrophoresis (SDS-PAGE). Subsequently, the proteins were transferred to a polyvinylidene fluoride membrane and blocked with 5% skim milk at 24 °C for 2 h. The membrane was then incubated with polyclonal antibodies against SPC (1:1000; AP53886PU-N; OriGene, Maryland, USA), HIF-1α (1:1000; AF1009; Affinity, Changzhou, China), EGFR (1:300; bs-10007R; Bioss, Beijing, China), EGF (1:300; bs2010R; Bioss Beijing China), PCNA (1:300; bs-2006R; Bioss Beijing China), Bax (1:300; bs-20386R; Bioss Beijing China), and β-actin (1:2000, Bioss, Beijing, China). After washing with PBS at 24 °C for 90 min, the membrane was further incubated with IgG antibody (1:3000, bs-0295G-HRP, Bioss, Beijing, China). Finally, an ECL luminescent solution was used for color development and the results were observed. β-actin served as the loading control. Grayscale values of protein bands were analyzed using ImageJ software to determine relative expression levels.

### 4.6. Isolation and Culture of Yak Alveolar Type II Epithelial (AT2) Cells

Lung tissue samples were collected from healthy yaks residing at an altitude of 2500 m. The lung tissue was prefaced with normal saline to remove any remaining blood. Blood vessels and connective tissue were avoided as much as possible to obtain alveolar tissue. The alveolar tissue was digested with 0.25% trypsin (27250018, Gibco, New York, NY, USA) at 37 °C for 45 min, and then digested with 0.1% (1 mg/mL) collagenase I (17100017, Gibco, New York, NY, USA) at 37 °C for 30 min. The cell suspension was filtered with 100 μm cell strainers, and the cells were cultured in 25 cm^2^ cell culture flasks in 5% CO_2_ and 21% O_2_ at 37 °C. After culturing for 40 min, the non-adherent alveolar epithelial cells were aspirated and centrifuged at 1000 rpm at 4 °C for 5 min, and the pellet was resuspended and inoculated in a new culture flask. This process was repeated 2–3 times, and the non-adherent cells in 10% fetal bovine serum DMEM/F12 (12400024, Gibco, New York, NY, USA) were inoculated into a new culture flask. If fibroblasts appeared after 4 d, AT2 cells were purified by 0.05% trypsin differential digestion. Fibroblasts were digested from the bottom of the flask much faster than AT2 cells. Cell passages were performed at a ratio of 1:2 or 1:3, and only 3–5 passages were used for subsequent experiments.

### 4.7. Transmission Electron Microscopy (TEM)

Lung tissue was prefixed with 3% glutaraldehyde, then postfixed in 1% osmium tetroxide, dehydrated in a series of acetone, infiltrated in EPON^®^ 812, and embeded. The semithin sections were stained with methylene blue and ultrathin sections were cut with a diamond knife, stained with uranyl acetate and lead citrate. The sections were examined using a JEM-1400-FLASH transmission electron microscope (JEOL, Tokyo, Japan).

### 4.8. Air-Liquid Interface (ALI) Cultures and Hypoxia

Primary AT2 cells were cultured in transwell plates with a 0.4 µm pore polyester membrane insert (3450, Corning, New York, NY, USA) as an air–liquid culture model. The normoxic group was cultured at 37 °C with 21% O_2_ and 5% CO_2_. Hypoxic conditions were achieved by culturing cells under hypoxic conditions of 5% O_2_ and 5% CO_2_ at 37 °C (Thermo Forma 3111, Niederelbert, Germany).

### 4.9. Cell Counting Kit-8 (CCK-8) Assays Cytotoxicity 

Cell Counting Kit-8 (CCK-8) was used to evaluate the cytotoxicity of DMSO (D2650, Sigma, St Louis, MO, USA) and ML228 (HY-12754, MedChemExpress, Junction, NJ, USA). Yak AT2 cells (5000 cells/well) were added to 96-well plates. After 24 h of culture in 37 °C with 5% CO_2_ and 21% O_2_, the basal medium was added for “starvation culture” for 12 h, and DMSO or ML228 were added at a specific concentration for 24 h. According to the instructions of the pharmaceutical factory, 10 μL CCK-8 was added to each well in a 96-well plate petri dish containing 100 μL medium. After incubation at 37 °C for 3 h, the absorbance at 490 nm was measured by a microplate reader. The results were compared with the relative optical density (OD) of untreated cells (defined as 100% survival).

### 4.10. 5-Ethynyl-20-Deoxyuridine (EdU) Assay 

AT2 cells were seeded into 6-well cell culture plates, and transfections were carried out once the cell density reached 30–40%. After 24 h of transfection, the cells were processed following the instructions of the Cell-LightTM EdU Apollo567 In Vitro Kit (RiboBio, Guangzhou, China). Subsequently, the cells were captured under a fluorescence microscope.

### 4.11. Statistical Analysis

All data are presented as mean ± standard deviation (SD) of independent experiments. Statistical analyses were performed using GraphPad Prism 6.0.0 for Windows (GraphPad Software, San Diego, CA, USA, www.graphpad.com, accessed on 12 October 2016) and SPSS 20.0 (IBM Corp. Released 2011. IBM SPSS Statistics for Windows, Version 20.0. Armonk, NY, USA: IBM Corp.). Comparisons were conducted using one-way analysis of variance (ANOVA), and *p* < 0.05, *p* < 0.01, and *p* < 0.001 were regarded as statistically significant and very significant.

## Figures and Tables

**Figure 1 ijms-25-01442-f001:**
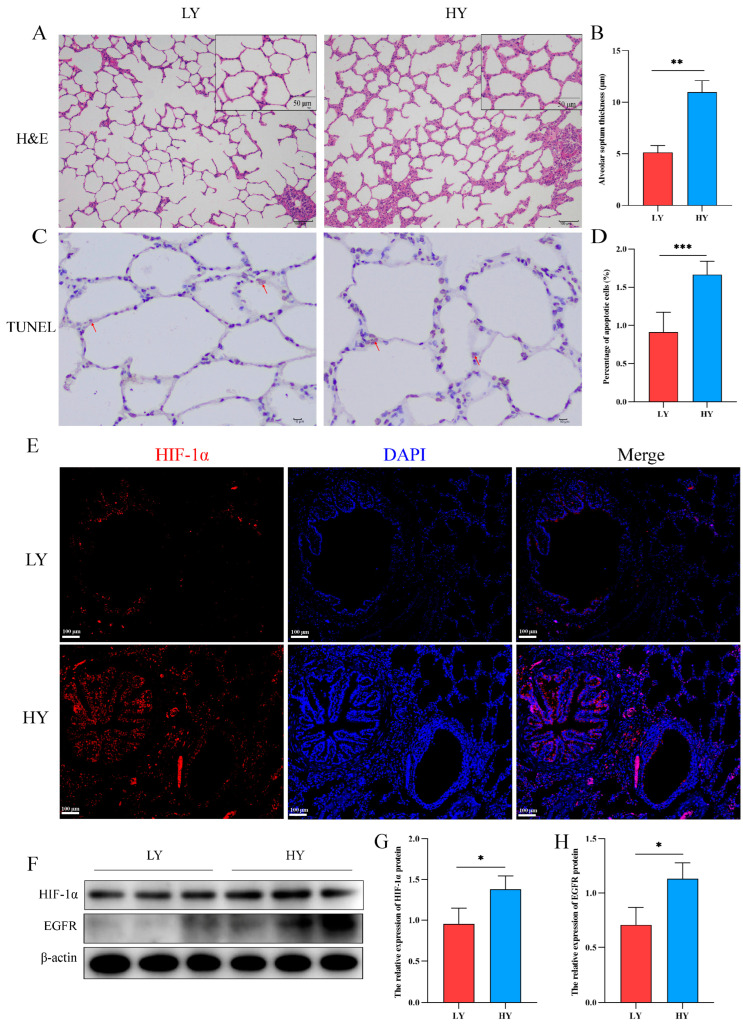
Histological structure and expression of HIF-1α/EGFR in the lungs of 6-month-old yak at different altitudes. (**A**) H&E staining was performed on the lungs of 6-month-old yaks at low altitude (LY) and high altitude (HY). (**B**) Statistical results of alveolar septum thickness (*n* = 6). (**C**,**D**) The TUNEL assay was utilized to detect apoptosis in alveolar epithelial cells and quantify the rate of cell apoptosis (*n* = 6). The apoptotic nuclei exhibit a light yellow or brownish yellow hue, while the normal nuclei display a blue or light blue coloration. The red arrows depict cells undergoing apoptosis. (**E**) Immunofluorescence was used to detect the distribution of HIF-1α in the lungs. (**F**–**H**) Western blotting was used to identify the expression variations of HIF-1α and EGFR in the lungs of 6-month-old yaks at varying altitudes (*n* = 3). Data are means ± SD. * *p* < 0.05, ** *p* < 0.01, *** *p* < 0.001.

**Figure 2 ijms-25-01442-f002:**
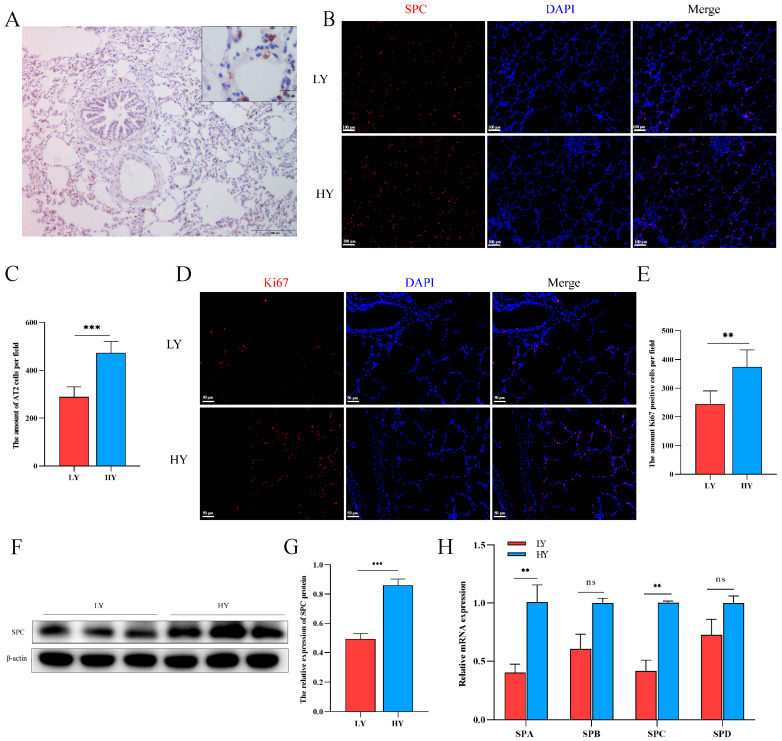
Effects of different altitudes on AT2 cell proliferation and alveolar surfactant-associated protein (SP) secretion in 6-month-old yaks. (**A**) The specific expression of SPC in the lungs of 6-month-old yaks was marked by immunohistochemistry. (**B**,**C**) Distribution of SPC and statistics of the number of SPC-positive cells using immunofluorescence in the lungs of 6-month-old yaks at different altitudes (*n* = 6). (**D**,**E**) Immunofluorescence detection of Ki67 distribution and statistics on the number of Ki67-positive cells in the lungs of 6-month-old yaks at different altitudes (*n* = 6). (**F**,**G**) Western blotting to detect differences in SPC expression in the lungs of 6-month-old yaks at different altitudes. (*n* = 3). (**H**) Expression patterns of SPA, SPB, SPC and SPD were detected by RT-qPCR. Data are means ± SD. ns stands for no difference, ** *p* < 0.01, *** *p* < 0.001.

**Figure 3 ijms-25-01442-f003:**
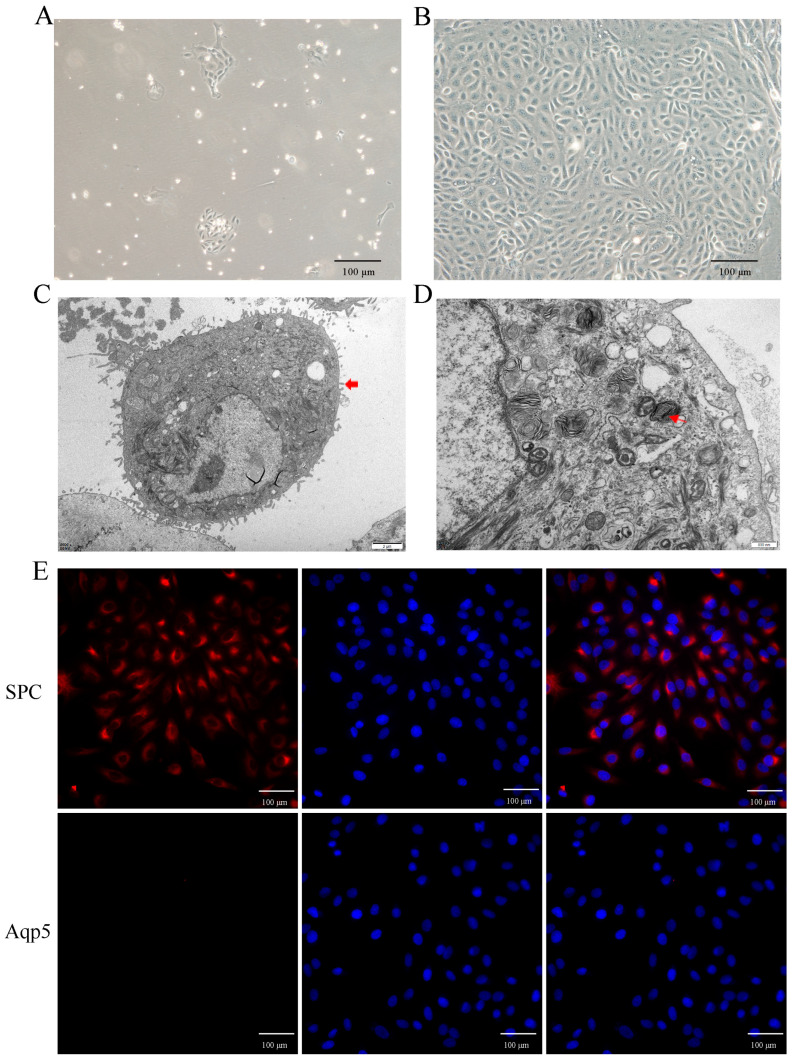
Isolation, culture, and characterization of yak primary AT2 cells. (**A**,**B**) The phase-contrast appearance of primary yak AT2 cells after culture in vitro for 72 h (**A**) and 6 days (**B**) The cultured cells had a stone-like appearance on the 6th day. (**C**,**D**) AT2 cells were identified using transmission electron microscopy. Thick red arrows indicate microvilli on the surface of AT2 cells, and thin red arrows indicate osmiophilic multilamellar bodies. (**E**) Identification of AT2 cells using positive staining with SPC (red) rather than Aqp5 (no expression, red). DAPI was used to stain the nucleus (blue).

**Figure 4 ijms-25-01442-f004:**
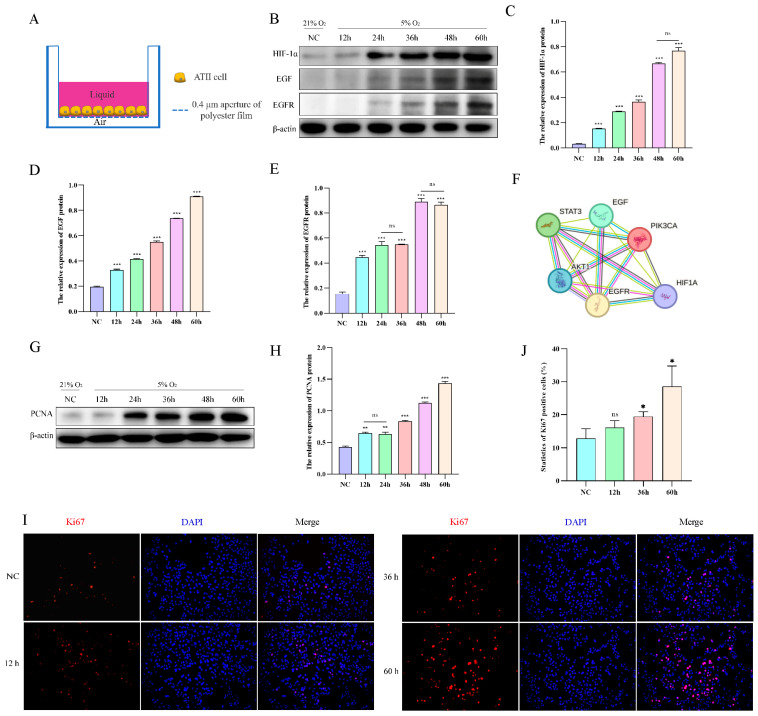
Effect of hypoxia on the proliferation of yak AT2 cells. (**A**) Diagram of the air-liquid culture model. (**B**–**E**) Protein expression of HIF-1α, EGF, and EGFR was measured after AT2 cells were exposed to normoxia or hypoxia for 12, 24, 36, 48, and 60 h. (*n* = 3). (**F**) The STRING database was used to retrieve the targeting relationship between HIF-1a and EGFR signaling pathway factors. (**G**,**H**) Protein expression of PCNA was measured after AT2 cells were exposed to normoxia or hypoxia for 12, 24, 36, 48, and 60 h. (*n* = 3). (**I**,**J**) After AT2 cells were exposed to normoxia or hypoxia for 12, 36, and 60 h, the expression of Ki67 and the number of positive cells were detected by immunofluorescence. (*n* = 6). Microscope magnification is 20×. Data are means ± SD. ns stands for no difference, * *p* < 0.05, ** *p* < 0.01, *** *p* < 0.001.

**Figure 5 ijms-25-01442-f005:**
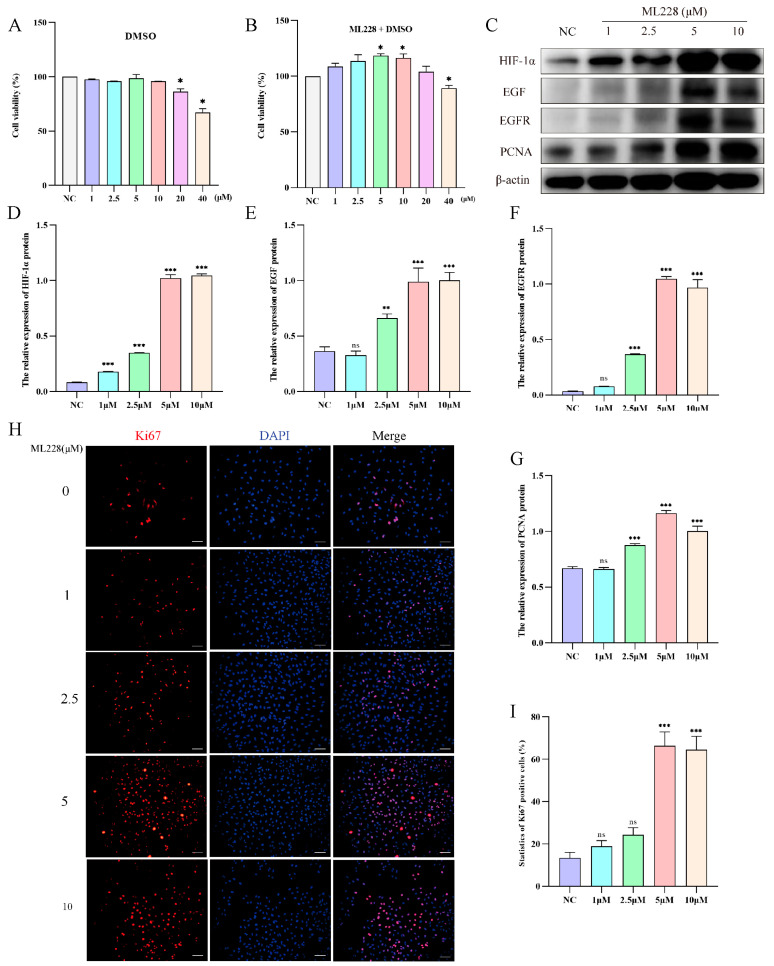
ML228 activates the HIF signaling pathway to enhance the proliferation of yak AT2 cells. (**A**,**B**) The cytotoxicity of DMOG and ML228 on AT2 cells was assessed using a CCK-8 assay. (**C**) Western blot analysis was performed to evaluate the impact of different concentrations of ML228 on the protein levels of HIF-1α, EGFR, EGF, and PCNA in AT2 cells. (**D**–**G**) Gray value analysis was conducted for Western blot results depicting expression levels of HIF-1α, EGFR, EGF, and PCNA proteins (*n* = 3). (**H**,**I**) Immunofluorescence staining revealed the quantification of Ki67-positive (red) cells in AT2 cells following treatment with varying concentrations of ML228 (*n* = 6). (bar = 100 μm). DAPI was used to stain the nucleus (blue). Data are means ± SD. ns stands for no difference, * *p* < 0.05, ** *p* < 0.01, *** *p* < 0.001.

**Figure 6 ijms-25-01442-f006:**
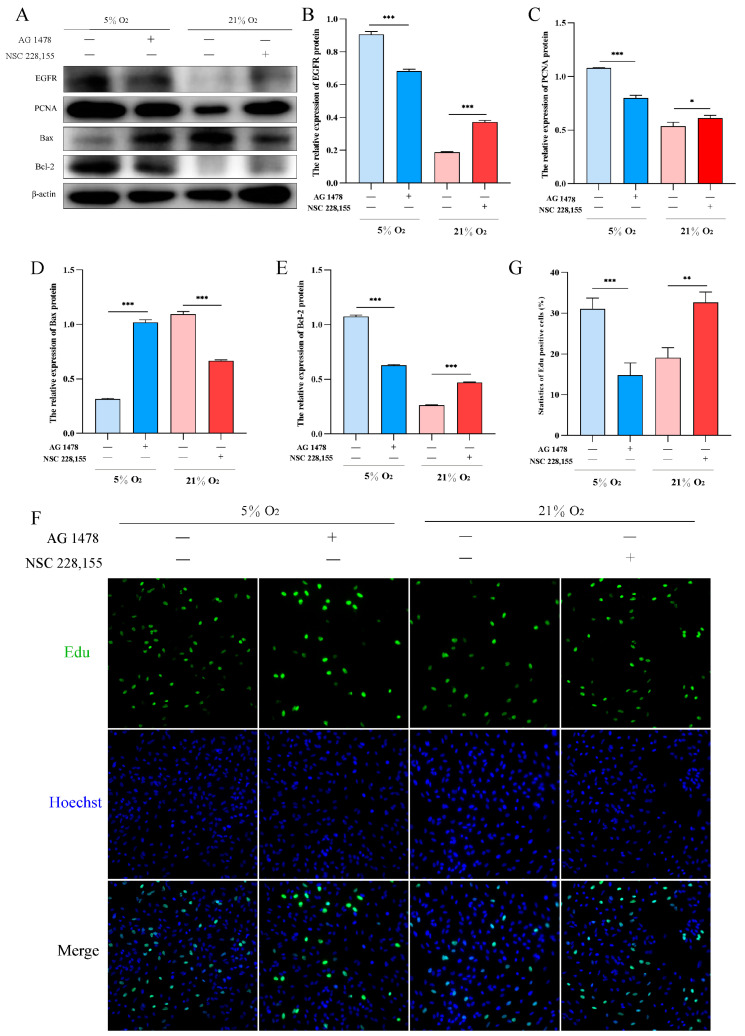
Effects of EGFR activation and inhibition on the proliferation and apoptosis of yak AT2 cells, respectively. (**A**) Western blotting was used to detect the expression of EGFR, PCNA, Bax, and Bcl-2 proteins after inhibiting and activating EGFR at 5% and 21% O_2_, respectively. (**B**–**E**) Gray value analysis of the Western blot of EGFR, PCNA, Bax and Bcl-2 expression. (*n* = 3). (**F**,**G**) Edu assay for AT2 cell proliferation after activation and inhibition of EGFR at 5% and 21% O_2_, respectively. (*n* = 6). The proliferating cells were labeled with EdU (green) and the nuclei were stained with Hoechst (blue). Microscope magnification is 20×.Data are means ± SD. * *p* < 0.05, ** *p* < 0.01, *** *p* < 0.001.

**Table 1 ijms-25-01442-t001:** Information on primers.

Genes	Primer Sequences (5’–3’)	TM/°C
*SPA*	F: 5′-GTCTTCTCTACCAATGGGCAGTCAG-3′ R: 5′-GGCAGCAATGTGTCCACCTACTC-3′	58
*SPB*	F: 5′-GCTGCCACAAGACTCTGACTGC-3′ R: 5′-CTCCTCCACAAACCGCTCACAC-3′	58
*SPC*	F: 5′-AAGATGGATGTGGGCAGCAAAGAG-3′ R: 5′-GTTCACGGGACAGCAAGGGATG-3′	58
*SPD*	F: 5′-GAGCATGAGCGACACCAGGAAG-3′ R: 5′-CACACAGTTCTCTGAGCCACCATC-3′	58
*β-actin*	F: 5′-TCCTGCGGCATTCACGAAACTAC-3′ R: 5′-GTGTTGGCGTAGAGGTCCTTGC-3′	58

## Data Availability

All data generated during the current study is included in this manuscript.

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
