# Peer review of "The HIF-1α/EGF/EGFR Signaling Pathway Facilitates the Proliferation of Yak Alveolar Type II Epithelial Cells in Hypoxic Conditions"

_ijms, 2024, doi:10.3390/ijms25031442_

Round 1
Reviewer 1 Report
Comments and Suggestions for Authors
Oxygen is a prerequisite for the survival of mammals, and the diverse oxygen conditions in different regions of China have resulted in the adaptation of unique species to the specific environmental conditions. One such representative animal is the yak. This study aims to investigate the changes in yaks and AT2 cells under different oxygen conditions using histological and cellular methods, with the hope of obtaining insights into the adaptation of yaks to oxygen. While this study has yielded some meaningful results, there are also several design flaws.
1. The origin of yaks from different altitudes needs to be clearly described. If yaks living at different altitudes have adapted to the local oxygen conditions over a long period of time, it is normal for them to experience lung damage when suddenly exposed to high altitudes. However, the information presented in Figure 1 suggests that long-term exposure to high altitudes leads to lung tissue damage and apoptosis in yaks. This finding is difficult to understand and requires a reasonable explanation.
2. In Figure 1E, the lung morphology of yaks from high altitude areas appears shrunken, which differs from the morphology shown in previous figures. Is this due to issues with sampling or tissue embedding?
3. The AT2 cells used in this study are derived from the lungs of yaks from low or high altitude areas. It is known that AT2 cells from yaks at different altitudes may exhibit differences in their response to oxygen. Therefore, subsequent experimental results, especially those obtained from high and low oxygen environments, may show significant discrepancies. In this study, two oxygen concentrations (5% and 20%) were chosen for testing. It is important to determine whether the oxygen concentrations in the cell culture medium match the actual concentration of oxygen in alveoli of these animals.
4. Different oxygen levels can affect many biological processes through HIF-1α. The HIF-1α/EGF/EGFR signaling pathway studied in this paper is involved in various aspects of lung epithelial in lung cancer. However, this study mainly focuses on cell phenotype data, with limited exploration of the specific mechanisms underlying HIF-1α under different oxygen conditions. This lack of innovation may be considered as a limitation.
Reviewer 2 Report
Comments and Suggestions for Authors
Manuscript ijms-2776025, entitled “The HIF-1α/EGF/EGFR Signaling Pathway Facilitates the Proliferation of Yak Alveolar Type II Epithelial Cells in Hypoxic Conditions”
Recommendation: The above paper is not suitable for publication in its present form.
This article provides useful information on the parameters affecting the proliferation of yak alveolar type II epithelial cells in hypoxic conditions. It is in general appropriately organized, carried out and written, however there are some points that should be corrected or clarified.
L33: “Due to its direct…”
L104: “shown” instead of “depicted”
In Fig. 4J, why are the “48h” missing?
L144: From 1 to 10 or 5-10μm?
L188: “…findings regarding the hypoxic…”
L189-190: “…reports in Tibetan pigs [12]. As indicated, these physiological…”
L200: “In this study, we also observed an…”
L207: “…attributed to their activity of replenishing…”
L209-210: “provides significant information” instead of “holds significance”
L253: 3+3?
L264: Please rephrase “Stain 4 μm thick sections on glass slides”
Comments on the Quality of English LanguageMinor editing of English language required
Round 2
Reviewer 1 Report
Comments and Suggestions for Authors
The author answered the questions of the reviewers by referring to the literature and the relevant results of this research group. No doubts at present.